# SARS-CoV-2 Spatiotemporal Genomic and Molecular Analysis of the First Wave of the COVID-19 Pandemic in Macaé, the Brazilian Capital of Oil

**DOI:** 10.3390/ijms231911497

**Published:** 2022-09-29

**Authors:** Bruno da-Costa-Rodrigues, Caio Cheohen, Felipe Sciammarella, Allan Pierre-Bonetti-Pozzobon, Lupis Ribeiro, José Luciano Nepomuceno-Silva, Marcio Medeiros, Flávia Mury, Cintia Monteiro-de-Barros, Cristiano Lazoski, Manuela Leal-da-Silva, Amilcar Tanuri, Rodrigo Nunes-da-Fonseca

**Affiliations:** 1Instituto de Biodiversidade e Sustentabilidade-NUPEM, Universidade Federal do Rio de Janeiro (UFRJ), Av. São José do Barreto 764, Macaé 27965-550, Brazil; 2Programa de Ciências Morfológicas, Instituto de Ciências Biomédicas, Universidade Federal do Rio de Janeiro (UFRJ), Rio de Janeiro 21941-170, Brazil; 3Instituto de Biologia, Universidade Federal do Rio de Janeiro (UFRJ), Rio de Janeiro 21941-902, Brazil

**Keywords:** SARS-CoV-2, COVID-19, epidemiology, Macaé, Brazil

## Abstract

The SARS-CoV-2 virus infection led to millions of deaths during the COVID-19 pandemic. Hundreds of workers from several other Brazilian cities, as well as from other countries, arrive daily in Macaé to work in the oil supply chain, making this city a putative hotspot for the introduction of new viral lineages. In this study, we performed a genomic survey of SARS-CoV-2 samples from Macaé during the first outbreak of COVID-19, combined with clinical data and a molecular integrative analysis. First, phylogenomic analyses showed a high occurrence of viral introduction events and the establishment of local transmissions in Macaé, including the ingression and spread of the B.1.1.28 lineage in the municipality from June to August 2020. Second, SARS-CoV-2 mutations were identified in patients with distinct levels of COVID-19 severity. Third, molecular interactions of the mutated spike protein from three B.1.1.33 local samples and human ACE2 showed higher interactions than that of the wild-type spike protein from the ancestral virus. Altogether, these results elucidate the SARS-CoV-2 genomic profile in a strategic Brazilian city and further explore the functional aspects of SARS-CoV-2 with a characterization of emerging viral mutations associated with clinical data and the potential targets for drug development against SARS-CoV-2.

## 1. Introduction

Severe acute respiratory syndrome coronavirus 2 (SARS-CoV-2) emerged in December 2019 in China and rapidly spread worldwide [1,2,3,4,5,6,7]. The first case was reported in Brazil in March 2020, and since then, the Brazilian health public authorities have faced serious challenges in controlling the coronavirus disease (COVID-19) pandemic, with more than 37,000,000 cases and 665,000 deaths registered by May 2022 [8].

The SARS-CoV-2 RNA genome possesses around 30,000 nucleotides, and understanding the genome is important for providing useful insights on the biology of this virus, including its mechanisms for infection, propagation, and pathogenesis [9,10,11]. As of August 2022, more than 12,000,000 SARS-CoV-2 genomes have been sequenced and made available on public databases, allowing researchers not only to investigate its biological features but also to generate RT-qPCR and antigen-based tools for rapid and precise diagnostics and vaccine production [12,13,14,15].

The ongoing evolution of SARS-CoV-2 strains is a global concern, and the monitoring and comprehension of SARS-CoV-2 genomic characteristics are essential for the development of strategies that target pandemic control. To track these specific problems, genomic surveillance efforts offer solutions that enable the analysis of its viral transmission and the discoveries of mutations that might have a direct impact on the clinical and epidemiological parameters of COVID-19 [1,15,16,17,18]. Additionally, SARS-CoV-2 sequencing provides valuable information about its evolution patterns, and important studies have focused on the understanding of its viral phylogenomics and lineage’s characterization [19,20,21,22]. 

In addition, a structural analysis between the SARS-CoV-2 proteins and human proteins, such as the S-Glycoprotein (spike protein) and human angiotensin-converting enzyme (hACE2), can guide the identification of new interaction hotspots and contribute to better defining the pathological profile of the infection [23,24]. A molecular modeling interaction analysis provides insights into the stability of the interactions that define the complexes interfaces and enables the discovery of possible new hotspots between the receptor-binding domain (RBD) from the spike protein and hACE2 [25,26]. Combining the SARS-CoV-2 genomic information and the analysis of its molecular and clinical potential outcomes is an important approach for a better comprehension of COVID-19’s epidemiological aspects [27,28,29].

Macaé is a coastal municipality located in the State of Rio de Janeiro and hosts over 50% of Brazilian oil production. Due to the daily arrivals of workers associated with the oil supply chain from different Brazilian cities and other countries, Macaé could rapidly become a Brazilian hotspot for the introduction of new variants of SARS-CoV-2. Therefore, this region was considered strategic for regional pandemic control.

Previously, our group provided detailed results of the SARS-CoV-2 distribution in Macaé and identified that COVID-19 cases in this city were male-biased, likely due to the prevalence of men as offshore workers [30]. Such studies of COVID-19 distribution in a local context are essential for understanding its macro effects on a specific region, enabling better decision-making for public health [30,31,32,33]. 

In this study, 96 SARS-CoV-2 positive samples from Macaé were selected for genome sequencing and molecular analyses. The samples were collected between April and August 2020 and comprised mild, severe illness (hospitalized), and critical (deceased) cases distributed across several Macaé neighborhoods. Four major outcomes were observed. First, a single B.1.1.33 SARS-CoV-2 lineage was identified in April 2020, while B.1.1.28 increased its frequency over the next few months. Second, phylogenomic analyses showed local clades of SARS-CoV-2 transmission in addition to the external introgression of new lineages. Third, three RT-qPCR positive samples were not detected by rapid antigen tests even with high similarity in their genome sequences, and their structural mutations were described. Fourth, different SARS-CoV-2 mutations were found according to the level of COVID-19 severity in patients, and molecular modeling indicated higher potential interactions of the mutant spike proteins with human ACE2 than that of the ancestral spike protein.

## 2. Results

### 2.1. Phylogenomic Analysis Revealed Local Transmission Clusters and Genomic Patterns Associated with the Distribution of Macaé SARS-CoV-2 Lineages

To investigate the genomic patterns of SARS-CoV-2 in Macaé, a maximum likelihood (ML) phylogenomic tree was built with the sequenced samples from Macaé, together with another 2743 deposited sequences from Brazil (downloaded from GISAID on 7 March 2021). From the 96 sequenced samples in this study, 76 samples passed the quality control (QC) step and were further analyzed with the other Brazilian sequences. We grouped 49 samples (64.48%) into 13 distinct clusters between April and August 2020. A total of 27 samples (35.52%) did not show any clustering pattern, suggesting independent introductions that did not contribute to further community transmission in Macaé (Figure 1A). Two of these clusters were characterized by a set of specific mutations that mainly occurred in samples from Macaé (Figure 1B). Seven samples from Macaé (9.2%) were grouped as cluster 1 and displayed a distinguished mutation of C4320T (orf1a:A1352V), while four samples (5.3%) constituted cluster 2 with the following common mutations: C3717T (ORF1a:A1151V), T10856C (ORF1a:E1293A), T12015G (ORF1a:V3917G), G23282T (S:D574Y), C25693T (ORF3a:L101F), and G29402T (N:D377Y). Samples from these two clusters were identified that dated between May and July 2020 and were highly associated with local transmission during this period, which contributed to the SARS-CoV-2 genomic profile in the city.

### 2.2. Macaé Displayed Punctual Introductions, Local Transmission, and a Lineage Frequency Variation of SARS-CoV-2 over Time during the Beginning of the COVID-19 Pandemic 

A spatiotemporal analysis of the SARS-CoV-2 viral cluster distribution in the urban area of Macaé during April–August 2020 was performed (Figure 2). A total of 13 sequenced samples did not show enough resolution to lineage characterization and were not further considered in this analysis. Eight other samples were obtained from residents outside the urban area of Macaé, and thus they are not represented in these maps. During April–May 2020, the B.1.1.33 lineage was the only clade found in Macaé, with 29 samples belonging to this clade. In April, the cluster 13 was the predominant one, while a diverse clustering pattern (1, 2, 5, 8, 9, 11, and 12) was observed in May. In June 2020, B.1.1.28 was detected for the first time, with its frequency representing 6% (*n* = 1) of analyzed samples, while B.1.1.33 remained as the major lineage, with 94% (*n* = 17) frequency. During this month, clusters 1, 5, and 8 had a higher frequency when compared with that of samples from clusters 6, 7, 12, and 13. Finally, the samples from July and August confirmed an increase in B.1.1.28 lineages, with 6% and 10% of B.1.1.28 (*n* = 2) in July and August, respectively, whereas B.1.1.33 remained the most predominant lineage with 94% and 90% (*n* = 36) in July and August, respectively. Cluster 4 was exclusively found in July and was the dominant cluster in that month, and cluster 12 appeared with higher frequency in August, with a widespread temporal distribution in this analysis and with its first report dated from May. These results highlight a dynamical evolution of the SARS-CoV-2 pandemics in Macaé, with large changes in the clustering profile between the two main viral lineages circulating in Brazil during this period. 

### 2.3. Different Mutations Are Associated with COVID-19 Severity

Tracking SARS-CoV-2 mutations is recognized worldwide as an important strategy for predicting infective patterns by viral-host canonical disturbances [34,35,36]. To correlate the physiological dynamics of the COVID-19 disease and its viral genetic mutations, SARS-CoV-2 genomes derived from Macaé patients were classified by severity. Patients were classified as mild (*n* = 33), hospitalized (*n* = 18), or deceased (*n* = 45) (see Material and Methods section for details). An analysis of the clinical data from those patients confirmed several previous reports comparing the clinical status and disease severity. Higher values of body mass index (BMI), lower saturation levels, and age were positively correlated with disease severity (Appendix A) [37,38]. 

A comparative sequence analysis of proteins N, M, and S with the SARS-CoV-2 Wuhan reference strain led to the identification of several mutations in the SARS-CoV-2 samples from Macaé between April and August 2020 (Appendix A). Twenty mutations on protein S, nine on protein N, and one on protein M were identified (Figure 3A,C). These mutations were investigated in relation to the degree of disease severity, and the mutations S:D614G, S:D574Y, and S:V1176F on the spike protein, together with the mutations N:R203K, N:I292T, and N:G204R on protein N, were identified in all groups, showing different levels of COVID-19 severity (Figure 3B,C). Furthermore, eleven unique mutations were associated with mild symptoms; two of those mutations were from hospitalized patients, and nine mutations were from deceased patients. 

During the development of this study, rapid antigen tests became available as an alternative to the RT-qPCR technology and as a fast and reliable approach for a SARS-CoV-2 diagnosis [39]. Twelve RT-qPCR positive samples circulating in Macaé between July and August 2020 that had their viral genomes sequenced in this study were also tested using rapid antigen tests. Six were positive (50%) in the antigen test, and six were negative (50%) (Appendix A). Three of the negative samples displayed RT-qPCR CTs of the N1 and N2 viral proteins in higher amounts or close to 30, suggesting that a lower amount of virus was present in the nasal swabs. The remaining three negative samples (65, 68, and 80) had viral targets CTs between 24 and 28, suggesting that enough virus was present in the swab sample for antigen detection. To test whether these false negatives were a result of a low viral load, a series of dilutions from one positive sample (86) were performed and showed that even a 1:64 dilution allowed for the detection of SARS-CoV-2 in the antigen test (Appendix A). A comparison of nucleocapsid (N) proteins obtained from 65, 68, and 80 genome samples did not show any mutations that could be correlated with the decrease in the sensitivity of the antigen rapid tests (Appendix A). 

### 2.4. Macaé Mutations in Spike Protein Led to Different Interaction Profiles with hACE2 by Molecular Dynamics

To estimate the impact of mutations derived from the viruses circulating only in Macaé, each one of the spike glycoprotein (S) mutated isoforms from cluster 2 were modeled. PSIPRED predicted a coiled structure for residues 1–25 and an alpha helix structure for residues 1150–1273. The chosen model showed a DOPE score of −392,875.15625, and a Ramachandran plot presented 89.1% of the modeled residues in most of the favored regions, 10.2% in additional allowed regions, 0.3% in generously allowed regions, and 0.4% in disallowed regions (Appendix A). A spike-hACE2 complex analysis was performed through molecular dynamics (MD) and provided intermolecular contacts, including hydrophobic and hydrogen bond interactions within the main hotspot in the RBD-SARS-CoV-2 interacting interface, R403, F497, Q498, T500, N501, G502, Y505, and Q506, against its counterpart hotspot in hACE2, which was formed mainly by residues E37, K353, G354, and D355 [26]. The spike proteins from three samples corresponding to the cluster 2 (Macaé-67, Macaé-70, and Macaé-80) were selected for a further analysis of the spike-hACE2 residue interactions (Figure 4A,B).

The electrostatic interactions between RBD and hACE2 (N501 and G354) were described as a hotspot residue. While the WT complex MD interaction occurred in only one out of the three runs (30%, 0%, and 0% of each simulation), Macaé-67 showed a higher frequency of interaction in every simulation (90%, 80%, and 60%) [26]. 

The T500 residue, present in the RBD domain from the SARS-CoV-2 spike protein, was defined as a hotspot for hACE2’s interaction with and shaping and determining of the stability of the protein–protein interface [40]. The interaction frequency of the WT protein in the MD analysis was always less than 40% of the simulation period (20%, 10%, and 40%), while the Macaé-70 sample showed hydrophobic interactions between T500 and L45 during the whole MD simulation (100%, 100%, and 100%). Other stronger hydrophobic interactions of the Macaé-70 sample MD when compared with that of the WT involved the residues Q506/K353. While this interaction was observed in only one of the three simulations in WT (0%, 0%, and 30%), the spike isoform from sample Macaé-70 had this interaction occurring in all of them (60%, 70%, and 80%). 

The Macaé-80 sample also showed a different interaction profile when compared with that of the WT in three residues: D405/Q388, F486/A80, and F486/M82. The interaction between D405 and Q388 was not previously described. This interaction did not occur in WT MD simulations more than 40% of the time (0%, 40%, and 10%), in contrast with the higher interaction of the Macaé-80 sample (90%, 60%, and 90%). D405 is conserved between SARS-CoV and the SARS-CoV-2 spike protein, and this residue has been suggested to play a role in RBD opening [29]. 

Hydrophobic interactions between F486 and A80 were found in all runs of Macaé-80 (70%, 70%, and 80%), while it significantly decreased in the wild type (30%, 0%, and 0%). The hydrophobic interaction that occurs between F486 and M82 is described as an interaction that helps to stabilize the RBD-hACE2 complex, and it was less observed in MD runs with the WT compared to that of Macaé-80 (30%, 10%, and 20% against 60%, 60%, and 80%, respectively) [11]. These data suggest an increase in the stability of the interaction between the S glycoprotein of SARS-CoV-2 and hACE2 in samples Macaé-67, Macaé-70, and Macaé-80, when compared with that of the WT SARS-CoV-2 strain.

## 3. Discussion

Since the beginning of the COVID-19 pandemic, Brazil has been one of the most affected countries, and until now, there are few analyses about SARS-CoV-2’s diversity characterization when compared with its continental territory [14,16,41,42,43,44,45,46,47,48,49]. Our results provide insightful information about its viral genomic characterization in the context of the COVID-19 pandemic’s evolution in an economically strategic Brazilian city. 

Our data highlight the tendency for the occurrence of different mutations in places where a continuous spread of the infection is closely accompanied by obstacles in epidemiological control. The identification of the lineage B.1.1.33 in the first few months of analysis is in agreement with the findings of national incidences, where B.1.1.33 and B.1.1.28 were the predominant Brazilian variants during this period [14,16]. Notably, the ongoing process of lineage evolution suggests a similar diversification process of the SARS-CoV-2 variants found in Macaé, where lineage B.1.1.28 has gradually increased its proportion over time. 

In a local context, our results show that punctual introductions were common during the period covered by this analysis, but local transmissions were also important for establishing the viral diversity observed in Macaé. Until May 2022, we did not find any reports in the literature regarding the identification or functional aspects of mutation orf1a:A1352V, which occurs in a region of the genome that is responsible for the production of non-structural proteins (NSPs). These proteins are reported as important players in SARS-CoV-2’s biology, with roles in viral transcription, replication, proteolytic processing, inhibition of host immune responses, and suppression of host gene expression [34]. The association observed for samples that carry this mutation with a higher mortality frequency suggests a possible link between these two variables and a potential increase in severity. However, more studies are necessary for a deeper understanding of these putative structure–pathogenesis connections. 

The mutations reported in this work reveal an unexpected local viral diversity, pointing out the necessity of strategic public policies for viral monitoring and pandemic control in cities with such special particularities, such as Macaé. One year after their original identification, mutations M:A98S and S:G842V still remain unobserved elsewhere in Brazil, while S:D614G underwent a rapid increase and fixation worldwide. Additionally, we reported several mutations with low frequency and poor or nonexistent functional studies that will hopefully contribute for further analyses and an understanding of SARS-CoV-2’s structural biology.

In this study, we sought to present an integrated approach combining genome sequencing and a clinical analysis of COVID-19 cases in Macaé. This approach enabled the characterization of different mutations in correlation with COVID-19 severity and provided evidence that supports the hypothesis of a relationship between mutation characteristics and test detection [50]. Although our study design found relevant mutations in correlation with the severity of COVID-19, further studies are necessary in order to increase the number of analyzed samples in both sequence and functional aspects. It is also interesting to note that, despite a false negative result on an Abbott rapid test, none of the samples possessed a mutation on the nucleocapsid, the viral protein detected by these antibody affinity tests. One plausible explanation for this observation is that mutations in regions that interact with protein N could induce conformational changes on the SARS-CoV-2 surface, leading to a decrease of its antigenic potential and culminating in the loss of the rapid test’s capability for detecting the virus. 

In addition, in silico analyses supply a snapshot of the spike-ACE2 interaction profile, which shows that mutations found in the city of Macaé may have a high affinity interaction between SARS-CoV-2’s RBD and hACE2, possibly impacting patient prognosis [51]. Furthermore, the substitution of residues, even spatially far from the hotspot on the spike RBD, was able to show changes in the ACE2 receptors’ residues. Future analyses that include an in vivo functional characterization of the described interactions, such as atomic force microscopy assays, will contribute to confirm our results in the empirical biology of COVID-19.

All strains analyzed held the spike D614G mutation. From the strains holding only this mutation, we saw that the residue V483 formed sustained hydrogen bonds with Q81 at 3.8 Å and hydrophobic interactions between E484 of the spike protein and F28 of ACE2. These residues belong to a loop region that extends further toward ACE2, establishing more extensive contacts with the receptor [52]. The electrostatic interaction N501/G354, the only WT discrepant found in sample Macaé-67, points to an enhanced ACE2 binding affinity [26].

D405 and Q388 residues present in the spike protein of the Macaé-80 sample perform hydrophobic interactions with ACE2, and the F486 residue interacts with both A80 and M82. Although F486 did not show interactions with WT during DM simulation, its importance in the RBD/hACE2 interaction was previously described [40,53]. Interactions with D405 also stand out for being facilitators in the RBD opening process, underscoring its importance for the interaction between the virus RBD region and the host cell [29]. 

Q506/K353 and T500/L45 hydrophobic interactions found in sample Macaé-70 show an ability of the virus to modulate its affinity for the hACE2 due to a robust number of times that both interactions occurred in simulation, leading to a greater capacity of infection in the host cell [54].

## 4. Material and Methods

### 4.1. SARS-CoV-2 RNA Extraction and RT-qPCR 

RT-qPCR tests were performed at the NUPEM-UFRJ Institute as previously described [19]. Briefly, nasopharyngeal swab samples from patients were stored in DMEM for RNA extraction with a MagMax kit (ThermoFisher, Waltham, MA, USA), where 200 μL of DMEM were used, following the manufacturer’s instructions. RNA samples were submitted to the Centers for Disease Control and Prevention (CDC)’s protocol for SARS-CoV-2 detection (Catalog # 2019-nCoVEUA-01), with GoTaq^®^ 1-step-RT-qPCR (Promega, Madison, WI, USA) and Taqman probes for N1, N2, and RNAseP (IDT-Integrated DNA Technologies, Coralville, IA, USA). The samples were considered positive only when two SARS-CoV-2 targets were amplified. 

### 4.2. Sample Selection for Genome Sequencing and Library Preparation

A total of 96 SARS-CoV-2 genomes were selected for third-generation sequencing. Nanopore technology fulfilled the following criteria: (1) samples were chosen from different neighborhoods of the city in order to allow for a broader genomic surveillance coverage; (2) samples were collected in equivalent proportions to represent a temporal analysis from April to August 2020; (3) samples were distributed according to adapted COVID-19 severity definitions by the National Institutes of Health (NIH). The COVID-19 treatment guidelines, for convenience purposes, were defined as mild (not needing any hospital care), severe illness (hospitalized), and critical (deceased). Details about the sample distribution are described in Appendix A. The cDNA library preparation for sequencing used the ARTIC-nCoV protocol [25,26]. Briefly, RNAs were reverse-transcribed to cDNA using SuperScript™ IV (ThermoFisher, Waltham, MA, USA), and PCR tests were performed using primer pools from ARTIC to retrieve 400 base pairs of PCR products. Next, we performed an end-repair on these products, and the barcodes were ligated using a NEBNext^®^ DNA Library Prep kit (NEB-New England Biolabs, Ipswich, MA, EUA) according to the manufacturer’s instructions. Finally, sequencing adapters were ligated using Quick T4 DNA ligase (NEB, Ipswich, MA, EUA), and the library was immediately loaded onto a Flongle™ flow cell on a MinION™ device (Oxford Nanopore Technologies, Oxford, UK). Twenty-four samples, each containing a unique barcode sequence, were loaded into each Flonge™ flow cell; four flow cells were used for the current analysis.

### 4.3. Post-Sequencing Processing and Bioinformatic Analysis

After sequencing, a high accuracy basecalling method using Guppy v3.2.10 (Oxford Nanopore Tech.) was applied. Read demultiplexing was enabled in MinKNOW, with the option “require_barcodes_both_ends” configured. A SQK-LSK109 kit and the barcoding expansion packs EXP-NBD104 and EXP-NBD114 were specified for all four runs. The sequencing run was monitored using RAMPART (https://artic.network/rampart (accessed on 10 November 2020)), and consensus sequences were generated using the medaka variation of the ARTIC-nCoV-bioinformaticsSOP-v1.1.0 pipeline (https://artic.network/ncov-2019/ncov2019-bioinformatics-sop.html (accessed on 16 November 2020). Samples where the number of ambiguous nucleotides that comprised more than 10% of the SARS-CoV-2 sequence (due to lack of coverage) were discarded from further processing. The percentage of bases with a coverage over 20x for each sequence are described in Appendix A. All genome samples were submitted to GISAID (https://www.gisaid.org/ (accessed on 19 July 2020)), and the respective IDs are described in Appendix A.

### 4.4. Lineage Identification and Sequence Analysis

The SARS-CoV-2 lineages were identified using the pangolin web application (https://pangolin.cog-uk.io/ (accessed on 19 March 2022)), with default parameters. The full-length genomic sequences of 2743 coronaviruses were aligned using the L-INS-i method of MAFFT v7.31023. The aligned sequences were converted into a PHYLIP file format using Clustal W. Maximum likelihood (ML) trees based on full-length genomic sequences were constructed and estimated with PhyML program version 3.025, using GTR+F+R5 model with 100 bootstrap pseudoreplicates. The evolutionary model and model parameters were estimated using the ModelFinder module from IQTree v2.1.2 [55,56]. The phylogenetic trees were visualized using FigTree v1.4.4 (http://tree.bio.ed.ac.uk/software/figtree/ (accessed on 12 September 2021)), and every genome sequence from Macaé was highlighted with a red dot in the tree. The samples that formed clusters were identified in the tree and categorized in numbers (Appendix A).

### 4.5. Geocodification of Each Viral Genome across Space and Time

Each sequenced sample was mapped into the municipality of Macaé using R 3.6.1. The Google Maps API was adopted from the R software package ggmap for geocoding as previously described [30], and samples were plotted every month (between April and August 2020). 

### 4.6. Rapid Antigen Test

Abbott Panbio Antigen rapid tests (code: 41FK10, lot: 41ADF016A; Abbott Laboratories, Chicago, IL, USA) were performed following the manufacturer’s instructions [39]. Briefly, 300 μL of the provided buffer were added to the extraction tubes with 50 μL of DMEM from the patients’ samples, and five droplets from this solution were added vertically in the specimen wells from the testing devices. After 15 min, the test results were registered to define positive or negative results. 

### 4.7. Molecular Modeling, Energy Minimization, and 3D Analyses

Three-dimensional structures were predicted by comparative protein modeling and consisted of the generation of 100 candidate models, where one model for each mutation was selected. We used the MODELLER v9.22 program and the SARS-CoV-2 spike protein bound to hACE2 as the template (PDBid: 7KMS). The secondary structures of the residues 1–25 and 1150–1273 were predicted using PSIPRED, built using the UCSF Chimera build structure function, and attached to the model using the Join Models function (200 ω angle, −130 φ angle). To introduce the desired mutations, the residues were manually changed using the Rotamers tool, and structure minimizations were performed for every model, with both steps performed on the UCSF Chimera [57].The validation of the stereochemical quality of the candidate models was performed using the DOPE (Discrete Optimized Protein Energy) score [58] and PROCHECK server Ramachandran plots [59]. The RMSD (root mean square deviation) between models and templates was determined using the UCSF Chimera software.

The structural flexibility and ten nanoseconds’ Molecular Dynamics (MD) simulations between the receptor-binding domain (RBD) of the SARS-CoV-2 spike protein and the hACE2 complex structures were performed and analyzed using the CABS-flex 2.0 server [31]. To ensure that results were not biased by a single structure, we performed the analysis in triplicate for each model, as well as with the wild type (WT) (PDBid: 7KMS). 

Molecular interactions between the complex chains were visualized with the MD Movie analyses module in the UCSF Chimera software, and a 4 Å distance cut-off was applied. LigPlot software was used to predict the hydrophobic and hydrogen interactions between both chains [60]. We retained the percentage of simulation time during which intermolecular contacts between hACE2 and SARS-CoV-2 spike protein RBD residues interacted in order to create an interaction profile. 

## 5. Conclusions

Our results contribute to a deeper understanding of how the pandemic arose and spread in such a strategic city for the Brazilian economy and a putative hotspot of new viral lineages. The genomic mutations analyses, allied with the clinical data reported in this study, might also contribute to a better understanding of the SARS-CoV-2 structure and function and its relationship to COVID-19’s severity and antigen rapid test detection. Our molecular modeling showed that protein S residues N501 and F486 might also be considered as potential targets for the development of new drugs against the coronavirus.

## Figures and Tables

**Figure 1 ijms-23-11497-f001:**
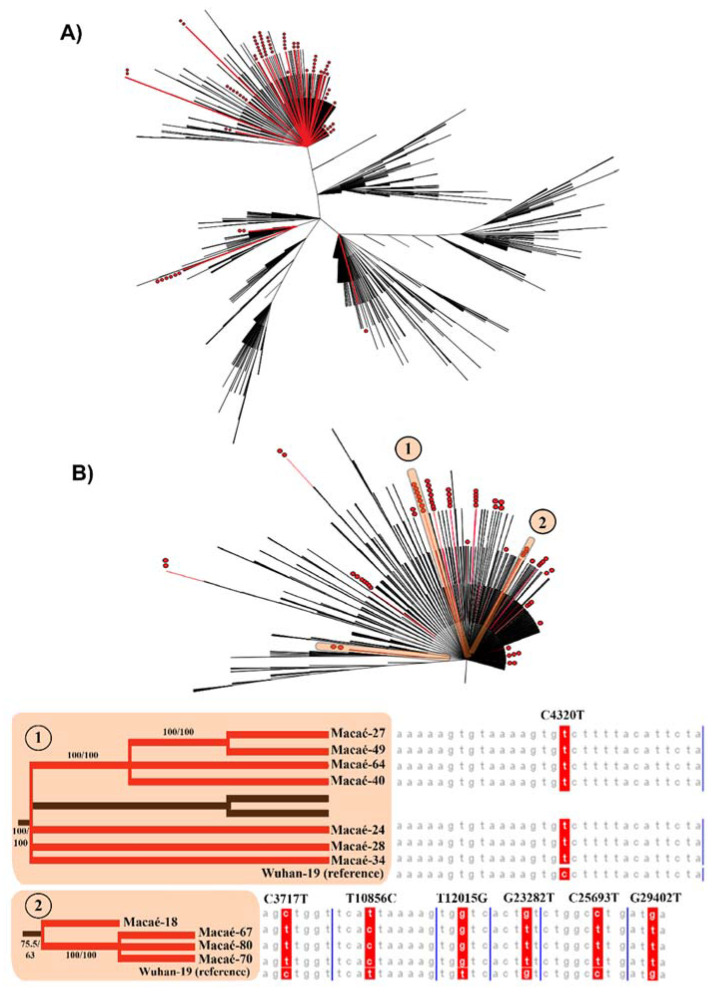
Phylogenomic analyses of SARS-CoV-2 variants show patterns of variant lineage insertion and local transmission in Macaé. (**A**) Phylogenomic relationship of 76 SARS-CoV-2 samples from Macaé (red branches) and from other regions of Brazil (black branches). Each red dot represents a sequenced sample generated in this study. (**B**) Phylogenomic trees from two Macaé clusters formed from local transmissions and their distinguished profile of mutations compared with that of the Wuhan-19 as the reference sequence on multiple sequences alignments. The branch support values are displayed above the respective branches and are represented as SH-aLRT/Bootstrap support values.

**Figure 2 ijms-23-11497-f002:**
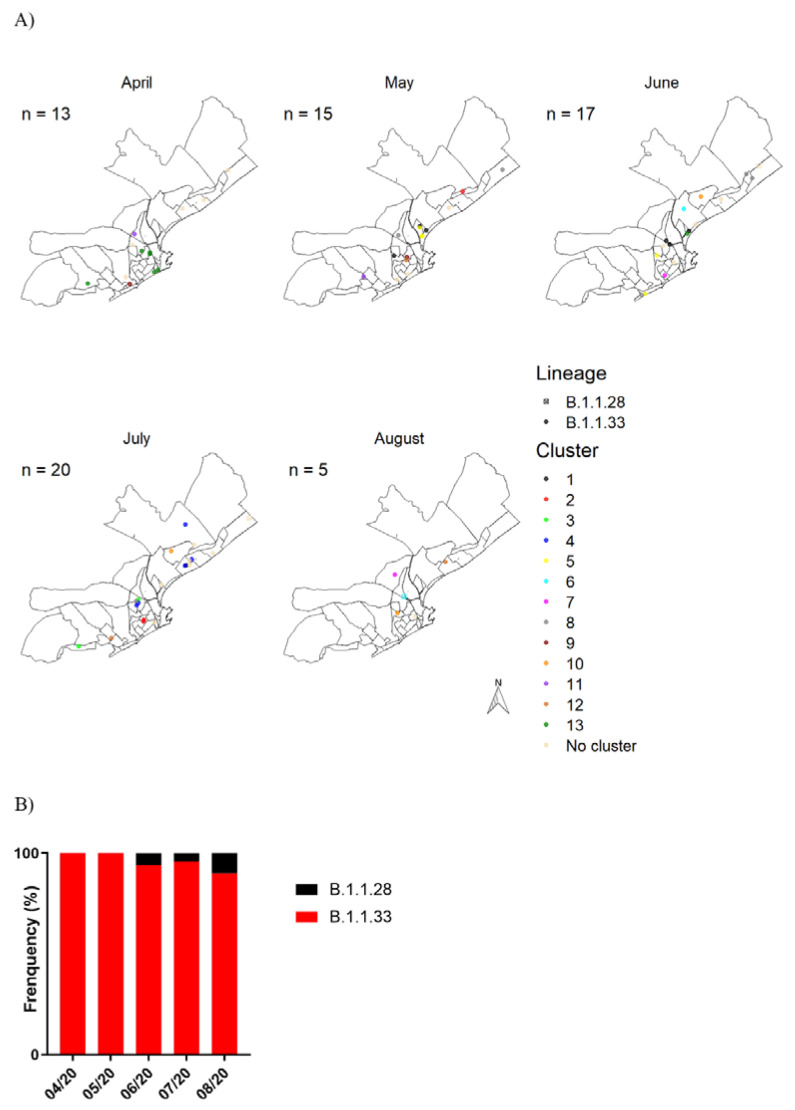
Spatiotemporal analysis of SARS-CoV-2 variants in the urban area of Macaé over the first few months of the pandemic in 2020. (**A**) Map of the urban area of Macaé showing the spatial representation of SARS-CoV-2 clusters and lineages over the first five months of the pandemic. (**B**) Histogram of the SARS-CoV-2 lineages’ frequency over the first five months of the pandemic.

**Figure 3 ijms-23-11497-f003:**
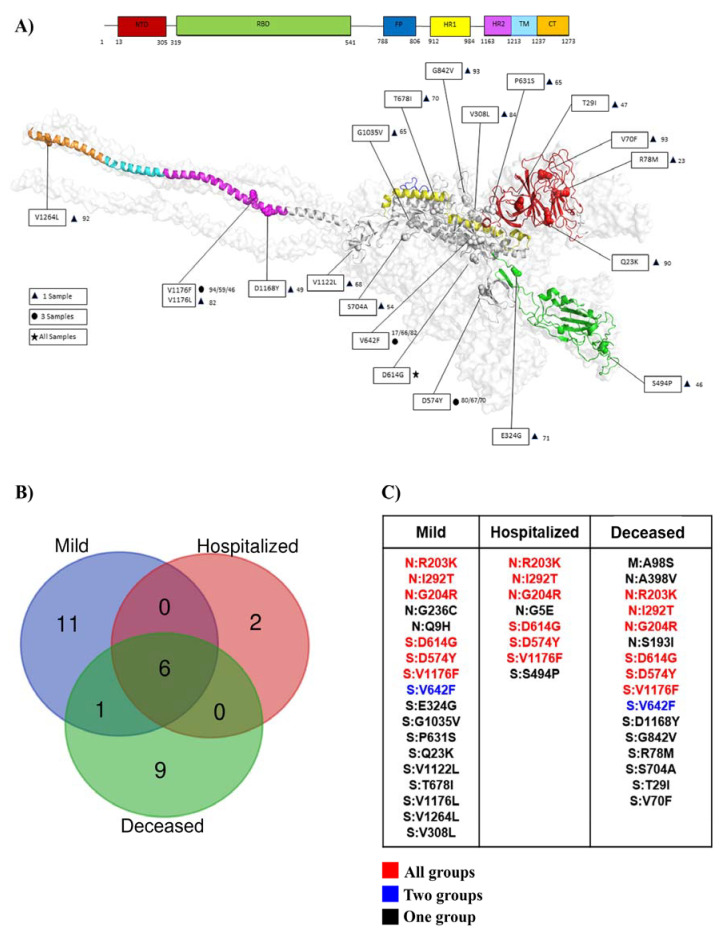
Comparative analyses of SARS-CoV-2 structural mutations in different levels of COVID-19 severity. (**A**) Structural profile of the SARS-CoV-2 spike protein, indicating all mutations identified in Macaé samples after sequencing. (**B**) Venn diagram indicating the correlation of structural mutations found in different levels of COVID-19 severity. (**C**) Table with a detailed description of the mutations in the SARS-CoV-2 structural proteins in different levels of COVID-19 severity. Shared mutations between the three groups are represented in red. Shared mutations between two groups are represented in blue. Exclusive mutations found in only one group are represented in black.

**Figure 4 ijms-23-11497-f004:**
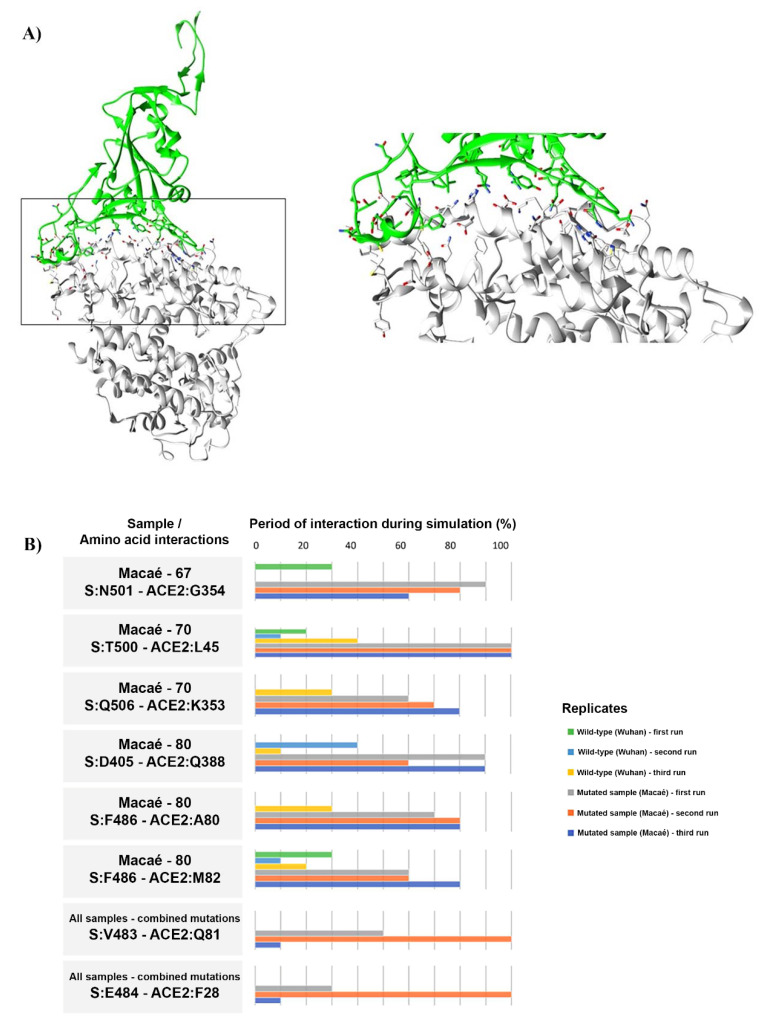
Molecular dynamics of the interaction between the spike protein and hACE2 show different profiles between the wild type and mutated spike isoforms from Macaé. (**A**) Molecular interaction profile from the spike protein of sample Macaé-70 with ACE2. Interactions between the spike protein and hACE2 are highlighted with a box and amplified for a more detailed view. (**B**) Histogram of the interaction profile from the wild type and the Macaé-mutated amino acids of the spike protein. Percentage number of simulations indicates the period that the specific residue had interacted with hACE2 during 10 nanoseconds of molecular dynamics.

## Data Availability

Detailed information from the SARS-CoV-2 sequenced samples and rapid antigen tests performed in this study can be found in Appendix A as an Excel file and at www.gisaid.org.

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
