# Peer review of "SARS-CoV-2 Spatiotemporal Genomic and Molecular Analysis of the First Wave of the COVID-19 Pandemic in Macaé, the Brazilian Capital of Oil"

_ijms, 2022, doi:10.3390/ijms231911497_

Round 1
Reviewer 1 Report
The manuscript, submitted by da-Costa-Rodrigues and colleagues for publication to Int. J. Mol. Sci., provided an in-depth analysis on the complete genome sequences, phylogenetic analysis and molecular modelling of 96 SARS-CoV-2 samples from Macaé, Rio de Janeiro, during the first outbreak of COVID-19, from April and August 2020. They carried out the exhaustive description of SARS-CoV-2 molecular epidemiology, along with an in silico representation in hACE2-Spike interaction in variants found in this place, supported by suitable statistical methods. The paper is well structured and exhaustive in every parts, while the methods is very interesting. In my opinion, it can be recommended for publication, upon addressing some minors.
In introduction, since you addressed the molecular interaction in silico, the authors should cite other experiences, where host-SARS-CoV-2 interaction was investigated with wider and explorative computational approach.
At pag 12 in Material and Methods, they did not report the mutation model used for ML tree, such as GTR or HKY, as it is basic parameter needed to finalize the phylogenetic analysis. Thus, they should report which model was used and explain how this model was chosen. Moreover, in Fig 1, especially in B, they should marked the nodes with bootstrap greatest than 85%, in order to test the genetic similarity described in these tree.
At pag 3, workflow for analysis on MinION output was reported, but did the authors take account a further correction step on reads, because of high error rate for single read (see 10.1371/journal.pone.0257521)?
The authors should add city and state to all companies (Promega; Themofisher).
In the title, Macaè is called as “Brazilian capital of oil”. It sounds as an informal way to indicate the sampling place of this study and, In my opinion, I would suggested to change it in “mining town” or “oil-rich city”, or also nothing.
Author Response
Dear reviewer,
We would like to thank you for your suggestions and valuable insights. Every critical suggestion were pertinent and contribute to the improvement of the manuscript. Below we address each one in order.
1 - In introduction, since you addressed the molecular interaction in silico, the authors should cite other experiences, where host-SARS-CoV-2 interaction was investigated with wider and explorative computational approach. I would suggest taking a look these papers of same research group: 10.1038/s41419-021-03881-8 and 10.1186/s12967-020-02405-w.
Reply: As suggested, we developed a paragraph to introduce computational approaches used for host-SARS-CoV-2 interaction analyses, which can be read at lines 43-49.
2- At pag 12 in Material and Methods, they did not report the mutation model used for ML tree, such as GTR or HKY, as it is basic parameter needed to finalize the phylogenetic analysis. Thus, they should report which model was used and explain how this model was chosen. Moreover, in Fig 1, especially in B, they should marked the nodes with bootstrap greatest than 85%, in order to test the genetic similarity described in these tree.
Reply: Again, the reviewer is correct, and we apologize to forget to detail this important information on methods. This was corrected in the newer version, as you can see at page 340. The evolutionary model used was inferred with IQTree version 2.1 with command line flag -m MFP, to perform model selection and parameter optimization. The final model selected was GTR+F+R5.
Also, we think that the description of branches support values, as well as the addition of the reference genome sequence for comparison of genetic similarity, suggested by reviewer 2, significantly increases the information of this phylogenomics analysis. Both suggestions were incorporated in fig 1b, and we really appreciate the respective suggestions.
3- At pag 3, workflow for analysis on MinION output was reported, but did the authors take account a further correction step on reads, because of high error rate for single read (see 10.1371/journal.pone.0257521)?
Reply: Regarding the correction and high error rates mentioned, the per-read high error-rate of the sequencer was mitigated with the use of the slow, but more accurate “High Accuracy” Model for basecalling. An analysis by Wick et. al (/10.1186/s13059-019-1727-y) in 2019 with an earlier version of the basecaller which, at the time, had a read accuracy of just above 87%, managed to provide a consensus accuracy over 99.37%. The
basecaller version in this paper got was released a few months after news from ONT that modal read accuracy was improved to around 96.2% and consensus accuracy to more than 99.99% (https://nanoporetech.com/about-us/news/new-research-algorithms-yield-accuracy-gains-nanopore-sequencing). Any high per-read error that may have remained would have been corrected by consensus and polishing steps downstream.
Furthermore, there was, indeed, a missing step in Materials and Methods. Post-basecalling sequences were assembled using the RAMPART workflow/software (https://artic.network/rampart), part of the ARTIC protocol. RAMPART is responsible for read counting, coverage calculation, primer trimming and assembly, as well as internally using Medaka (https://github.com/nanoporetech/medaka), a tool provided by Oxford Nanopore Technologies to perform consensus sequence polishing through a trained Neural Network Models utilizing the pre-assembly reads as input. These details were added to section “4.3 Post-sequencing processing and bioinformatic analysis”.
4- The authors should add city and state to all companies (Promega; Themofisher).
Reply: We thank the reviewer for point this detail, and we have correct those omissions along the paper.
5- In the title, Macaé is called as “Brazilian capital of oil”. It sounds as an informal way to indicate the sampling place of this study and, In my opinion, I would suggested to change it in “mining town” or “oil-rich city”, or also nothing.
Reply: We thank the reviewer for this suggestion. However, we would appreciate to maintain this description in the title, for the reasons that Brazilian capital of oil is how Macaé is internationally known, and to be in accordance with our previous report of COVID-19 characteristics in the city (see 10.1038/s41598-021-99475-7). We have made a small change in the title to contemplate all described on this reply.

Reviewer 2 Report
1. Line 38: “To date” should be replaced with “As of [a date]” to address time lag between manuscript submission and publication.
2. Figure 1: (a) It is very difficult to identify red branches on panels A and B, partly due to the blue shade; (b) it would be very helpful to add reference sequence(s) to give readers more information about genetic diversity present; (c) what do dots at nodes refer? and (d) legend on the two multiple sequence alignments appears to be missing.
3. Discussion section should end with study limitations and concluding remarks (aka take home messages), which are both missing in the current version. Suggest consolidating the “conclusion” section into “Discussion”.
4. Authors classification of mild (non-hospitalised), severe (hospitalized but survived) and critical (dead) covid-19 cases is different from definitions of WHO and other major health agencies such as the US CDC. More background information should be provided if the classification is used nationwide or just in the current study for convenience purpose.
5. The authors identified mutation patterns that may be associated with spike affinity to ACE2 receptor and thus to clinical severity. One statistical analysis for the authors’ consideration is to perform Cox regression modelling to identify any specific mutation and mutation panel that can significantly predict clinical severity outcome. This analysis is not essential to the current study but the authors should seriously consider such a clinically-relevant analysis in the future. To this reviewer, the outcome of Cox analysis is far more impactful than in silico analysis currently presented.
Author Response
Dear reviewer,
We would like to thank you for your commitment on analyze, correct and improve this manuscript. We took all requests into consideration, and respond as follows:
1. Line 38: “To date” should be replaced with “As of [a date]” to address time lag between manuscript submission and publication.
Reply: We thank the reviewer for notice this sentence. The corrected sentence can now be seen in line 38.
2. Figure 1: (a) It is very difficult to identify red branches on panels A and B, partly due to the blue shade; (b) it would be very helpful to add reference sequence(s) to give readers more information about genetic diversity present; (c) what do dots at nodes refer? and (d) legend on the two multiple sequence alignments appears to be missing.
Reply:
a) After the reviewer’s consideration, we did agree that the blue shade could cause difficulties for some reader’s visualization, and for this reason we remove it from the figure in order to highlight the red branches.
b) We think the reviewer has a fair point and is a good ideia to add the reference sequence. We have add it in Fig.1b in conjunction with the branches support values, suggested by reviewer 1, and we believe it significantly improved the information of this figure.
c-d) Again we thank the reviewer for point these omissions. We have corrected these mistakes, and all proper information and details regarding these observations can now be seen at figure’s 1 legend.
3. Discussion section should end with study limitations and concluding remarks (aka take home messages), which are both missing in the current version. Suggest consolidating the “conclusion” section into “Discussion”.
Reply: We think the reviewer’s point increase the quality and understanding of the manuscript, so we have added the study limitations into discussion. We did not changed the conclusions section in order to preserve the manuscript structure requested by the journal for author’s submission, and we also believe this structure highlight the take home messages, as suggested by reviewer.
4. Authors classification of mild (non-hospitalised), severe (hospitalized but survived) and critical (dead) covid-19 cases is different from definitions of WHO and other major health agencies such as the US CDC. More background information should be provided if the classification is used nationwide or just in the current study for convenience purpose.
Reply: We thank the reviewer for the critical observation, and believe this important information should indeed be more detailed. We have improved this information in material and methods, which can be seen in line 311.
5. The authors identified mutation patterns that may be associated with spike affinity to ACE2 receptor and thus to clinical severity. One statistical analysis for the authors’ consideration is to perform Cox regression modelling to identify any specific mutation and mutation panel that can significantly predict clinical severity outcome. This analysis is not essential to the current study but the authors should seriously consider such a clinically-relevant analysis in the future. To this reviewer, the outcome of Cox analysis is far more impactful than in silico analysis currently presented.
Reply: The authors acknowledge the valuable indication of Cox regression analysis to identify any specific mutation and mutation panel that can significantly predict clinical severity outcome. However, the Cox regression model allows the analysis of data in which the time until the occurrence of an event of interest is modeled, adjusted for covariates. As our study was not initially designed to contemplate this outcome, this methodology could not be adopted at this manuscript, but we surely will take this consideration for our future works.

Reviewer 3 Report
The authors present genomic analysis results of SARS-CoV-2 cases in Macae, Brazil. Genomic epidemiology is a useful method for identifying the underlying epidemiological progress of disease outbreaks. However, this study analyzed only 76 samples, which is not sufficient to make a scientific conclusion. For example, the relationship between mutation and disease severity should be analyzed using more big data. In addition, the study was conducted for the small geographic region, Macae. The scientific significance of this looks low because the results were obtained from a restricted number of samples and a small geographic area. More than 10 million sequences are available in GISAID, and other studies using big genomic data are continuously published now. A more large-scale analysis, such as in Brazil, will be needed for publishing data in international journals. A Brazilian local journal seems more appropriate for publishing this data.
Author Response
The authors present genomic analysis results of SARS-CoV-2 cases in Macae, Brazil. Genomic epidemiology is a useful method for identifying the underlying epidemiological progress of disease outbreaks. However, this study analyzed only 76 samples, which is not sufficient to make a scientific conclusion. For example, the relationship between mutation and disease severity should be analyzed using more big data. In addition, the study was conducted for the small geographic region, Macae. The scientific significance of this looks low because the results were obtained from a restricted number of samples and a small geographic area. More than 10 million sequences are available in GISAID, and other studies using big genomic data are continuously published now. A more large-scale analysis, such as in Brazil, will be needed for publishing data in international journals. A Brazilian local journal seems more appropriate for publishing this data.
Reply:
Dear reviewer,
We would like first to thank you for your time on analyzing and sharing your comments regarding this manuscript. We do agree with the reviewer that, in general, the higher the number of any subject of study, the higher it is the statistical power from the analysis. That said, it is important to restate that the goal of this study was to perform an integrated epidemiological analysis of a strategical city from Brazil, which contemplate several different approaches that we would like to briefly highlight:
• The choice for epidemiological analysis in Macaé was strategical, as the city is one of the main points of entrance for foreigners in the country due to its international nature of oil exploration. The understanding of epidemiological aspect in places such as Macaé, Wuhan, and others, is crucial for the understating of the pandemics at a larger scale, as well as deeply contributes for countries’ effective public policies. In fact, we are happy to say that the integrative analysis reported in this manuscript, together with our previous report (see 10.1038/s41598-021-99475-7), was responsible for save hundreds of lives in the city of Macaé, and contributed in made this city to possess the lower mortality rate in the state of Rio de Janeiro, the third most populous state of Brazil. The outstanding performance of Macaé during pandemics in one of the most affected countries by COVID-19 in the world highlights the relevance of this initiative, and such approach and results should be shared with the international scientific community.
• From the 96 sequenced samples from Macaé performed on this study, 91 sequences had supported quality for insertion at GISAID database, which represents today 40% of all sequences from this region annotated on such database. In addition, the study was designed for contemplate every places of the city, which gave us the broadest dimension for epidemiological analysis with the resources that were available in such period, which allowed us to infer in a temporal and genomic perspective, the viral lineages and events of insertion and clusterization that have occurred in the city. Also, we have combined those sequences information with every single clinical data from the respective hosts that were available, increasing the complexity of this study and improving the
manuscript with insightful details. Finally, important reports in the literature that comprises mainly SARS-CoV-2 genome sequencing features possess similar or lower (sequenced samples/population number) ratio from this study. Just in brief, see 10.1038/s41591-020-1000-7, 10.1186/s12864-021-07708-w, 10.1038/s41467-021-25985-7, 10.1038/s41467-021-25137-x, 10.1038/s41467-022-29579-9, 10.1038/s41467-020-20688-x.
• Combined, the annotation of the sequenced samples provided 125 characterized mutations, being twenty found in the spike protein genomic region, 9 in the nucleocapsid, and 1 in the membrane.
• We excluded 20 sequences from the initial 96 sequenced samples on the phylogenomics analysis in order to perform the best practices and include only the sequences that had the highest support by quality control and statistical criteria. Combined with those sequences, we performed the phylogenomics analysis together with other 2.743 Brazilian SARS-CoV-2 sequences from the same period, and by that, we analyzed the sequences profile from Macaé’s samples in a national context, and observed the points of introductions and local transmissions in the city.
• We performed an in silico molecular dynamics in order to analyze the qualitative and quantitative profile of interactions between mutated spike samples found in Macaé and human ACE2.
Altogether, this study comprises an extensive epidemiological analysis that comprises different approaches and is supported by referenced statistical methods, and therefore, should be taken into consideration as it possesses relevant outcomes.
In order to attend all three reviewer’s concerns and considerations, we have significantly improved this manuscript according to their requests as follows:
- We have included more background and literature for a better understand of our study design and findings (as it can be seen in line 43), and also corrected some minors throughout the text regarding misspellings.
- Our material and methods section was also modified, in order to increase the level of details on several important information for a better comprehension of the study design. Those modifications can now be seen throughout the material and methods section, where all sections had substantial improvements.
- We improved the presentation of our results, modifying figure 1 for increase the quality of visualization, and also detailed branches support values and legend, as it can be seen in page 3.
- We improved the statement of results, detailing every statistical measures and study design, and added the limitations from this study in the discussion section, which includes this reviewer concern regarding the number of patients in mutations analysis.
We thank all reviewers for their efforts in improve this manuscript, and we send attached the new version from this manuscript for appreciation, which contains every requests contemplated.
Kind regards,
